

# Evaluating the cleavage efficacy of CRISPR-Cas9 sgRNAs targeting ineffective regions of *Arabidopsis thaliana* genome

Afsheen Malik[1], Alvina Gul[1], Faiza Munir[1], Rabia Amir[1], Hadi Alipour[2], Mustafeez Mujtaba Babar[3], Syeda Marriam Bakhtiar[4], Rehan Zafar Paracha[5], Zoya Khalid[6] and Muhammad Qasim Hayat[1]

[1] Department of Plant Biotechnology, Atta-ur-Rahman School of Applied Biosciences, National University of Sciences and Technology, Islamabad, Pakistan
[2] Department of Plant Production and Genetics, Faculty of Agriculture and Natural Resources, Urmia University, Urmia, Iran
[3] Shifa College of Pharmaceutical Sciences, Shifa Tameer-e-Millat University, Islamabad, Pakistan
[4] Department of Bioinformatics and Biosciences, Capital University of Science and Technology, Islamabad, Pakistan
[5] Research Center for Modeling and Simulation, National University of Sciences and Technology, Islamabad, Pakistan
[6] Computational Biology Research Lab, Department of Computer Science, National University of Computer and Emerging Sciences-FAST, Islamabad, Pakistan

Corresponding author
Alvina Gul, alvina_gul@asab.nust.edu.pk

## ABSTRACT

The CRISPR-Cas9 system has recently evolved as a powerful mutagenic tool for targeted genome editing. The impeccable functioning of the system depends on the optimal design of single guide RNAs (sgRNAs) that mainly involves sgRNA specificity and on-target cleavage efficacy. Several research groups have designed algorithms and models, trained on mammalian genomes, for predicting sgRNAs cleavage efficacy. These models are also implemented in most plant sgRNA design tools due to the lack of on-target cleavage efficacy studies in plants. However, one of the major drawbacks is that almost all of these models are biased for considering only coding regions of the DNA while excluding ineffective regions, which are of immense importance in functional genomics studies especially for plants, thus making prediction less reliable. In the present study, we evaluate the on-target cleavage efficacy of experimentally validated sgRNAs designed against diverse ineffective regions of *Arabidopsis thaliana* genome using various statistical tests. We show that nucleotide preference in protospacer adjacent motif (PAM) proximal region, GC content in the PAM proximal seed region, intact RAR and 3rd stem loop structures, and free accessibility of nucleotides in seed and tracrRNA regions of sgRNAs are important determinants associated with their high on-target cleavage efficacy. Thus, our study describes the features important for plant sgRNAs high on-target cleavage efficacy against ineffective genomic regions previously shown to give rise to ineffective sgRNAs. Moreover, it suggests the need of developing an elaborative plant-specific sgRNA design model considering the entire genomic landscape including ineffective regions for enabling highly efficient genome editing without wasting time and experimental resources.

## INTRODUCTION

Traditionally, scientists have been using various physical, chemical and biological techniques like irradiation, chemical and insertional mutagenesis either for incorporating traits of agricultural importance in crop plants or studying and deciphering important biological mechanisms in model plants (*Wu et al., 2005*; *Tadege et al., 2009*; *Oladosu et al., 2016*). However, the potential disadvantages associated with these traditional techniques are that all of these methods induce mutations in genome randomly that have a high tendency of producing undesired mutations and phenotypes (*Shalem, Sanjana & Zhang, 2015*; *Chaudhary et al., 2019*). Moreover, search for desired mutations requires screening bulk populations often accompanied by constructing mapping population and map-based cloning which are laborious, costly, and time-consuming processes (*Gilchrist & Haughn, 2010*; *Lee, Gould & Stinchcombe, 2014*). Thus, the development of techniques that can transform plant genetics and improve crop plants by overcoming these limitations are highly desired. The discovery of designer nucleases, which can be engineered for targeted genome editing, has emerged as a powerful tool over current approaches (*Rinaldo & Ayliffe, 2015*). Among these nucleases, Clustered Regularly Interspaced Short Palindromic Repeats (CRISPR)-CRISPR-associated (Cas) system has evolved as a simpler and efficient mutagenic tool that can be used in diverse organisms including plants (*Sander & Joung, 2014*; *Hussain, Lucas & Budak, 2018*). The CRISPR-Cas system works as a part of the bacterial or archaeal adaptive immune system where it safeguards them from invading foreign DNA molecules (*Barrangou et al., 2007*; *Wiedenheft, Sternberg & Doudna, 2012*). The standard CRISPR-Cas9 system, derived from *Streptococcus pyogenes*, is widely adopted for mediating targeted genome editing due to its relative simplicity. In this regard, a major breakthrough occurred when synthetic chimera of CRISPR RNA (crRNA) and trans-activating crRNA (tracrRNA) moieties known as single guide RNA (sgRNA) was generated that successfully guided the Cas9 to specific sites in the genome for targeted editing (*Jinek et al., 2012*). The 20-nucleotide spacer sequence (denoted as gRNA in the current study) at the 5′ end of the sgRNA directs the Cas9 protein to the complementary target sequence marked by NGG protospacer adjacent motif (PAM) present downstream of it for inducing double stranded breaks (DSBs) and also determines the specificity and cleavage efficacy of Cas9 endonuclease (*Jinek et al., 2012*; *Wong, Liu & Wang, 2015*).

Despite the simplicity and robustness of the system, the sgRNA specificity and on-target cleavage efficacy are the major concerns in CRISPR-Cas9 mediated genome editing. Different computational tools have been developed for determining the sgRNAs specificities (reviewed in *Liu, Zhang & Zhang, 2019*). Moreover, double nicking and transcriptional activation domain-based studies have been shown promising for improving sgRNA on-target specificity (*Ran et al., 2013*; *Mali et al., 2013*). Determining the sgRNA specificity and off-target prediction for mammalian systems is very important compared to plants, as the backcrossing can easily alleviate off-target effects in plants (*Kim, Alptekin & Budak, 2018*; *Naim et al., 2020*). The second important factor determining the Cas9 effectiveness that impacts both mammalian and plant systems is the on-target cleavage efficacy of sgRNAs. Recently, sgRNAs cleavage efficacies have been realized, as several groups have

identified various sequence and structural features of sgRNAs affecting their on-target cleavage efficacy and have developed models and algorithms, which are now incorporated in different computational tools for designing optimum sgRNAs (*Cong et al., 2013*; *Doench et al., 2014*; *Heigwer, Kerr & Boutros, 2014*; *Wang et al., 2014*; *Wu et al., 2014*; *Xie, Zhang & Yang, 2014*; *Chari et al., 2015*; *Fusi et al., 2015*; *Housden et al., 2015*; *Moreno-Mateos et al., 2015*; *Wong, Liu & Wang, 2015*; *Xu et al., 2015*; *Doench et al., 2016*; *Liang et al., 2016*; *Cao et al., 2017*; *Chari et al., 2017*; *Labuhn et al., 2018*; *Mendoza & Trinh, 2018*; *Labun et al., 2019*). Despite all these advancements, some major issues are associated with almost all of these models. For instance, models are trained on datasets derived from few mammalian systems. Datasets are derived from coding regions of genomes that add biasness to the analysis and models. Moreover, not all of these tools have a user-friendly interface and there is a lack of consistency among the outputs, which raises reliability concerns (*Liu, Zhang & Zhang, 2019*). On the other hand, in the plant science community, the problem is more complicated by the fact that only some plant-specific sgRNA prediction tools are available that offer sgRNA design for a limited number of plant genomes (Table 1). Most of the plant sgRNA design tools use mammalian systems derived models for off-target predictions and determining on-target efficacy, thus giving rise to inconsistency and discrepancies between predicted and observed in vivo CRISPR-Cas9 working. Furthermore, studies evaluating the on-target cleavage efficacies are lacking in plants (*Liang et al., 2016*; *Naim et al., 2020*). Thus, all these factors demand further work in these directions.

The non-coding regions of DNA not only maintain the structure of chromatin but also harbor important regulatory elements (*Böhmdorfer & Wierzbicki, 2015*; *Shanmugam, Nagarajan & Pramanayagam, 2017*). Almost all sgRNA design models and algorithms are trained on datasets that exclude non-coding regions from analysis because of their potential to give rise to non-effective sgRNAs despite realizing the importance of these regions. Therefore, most plant functional genomics studies that require the deletion of large chromosomal parts or deciphering the functional role of regulatory elements often face failure because of the inability of sgRNA design tools for predicting efficient sgRNAs against these regions (*Durr et al., 2018*). Thus, for the successful application of CRISPR-Cas9 technology against non-coding regions, consideration of these regions is of immense importance while orchestrating the models.

In the present study, we target DNA regions that are excluded from sgRNA design model (*Doench et al., 2014*) along with other non-coding regions for determining the various sequence and structural features of sgRNAs potentially associated with their high on-target cleavage efficacies against these regions. These regions include 5′ untranslated regions (5′ UTRs), 3′ untranslated regions (3′ UTRs), introns, area near N- and C-terminal regions, which were reported as "broadly ineffective target regions" for giving rise to ineffective sgRNAs (*Doench et al., 2014*), long non-coding RNAs (lncRNAs) and intergenic regions (hereafter all these regions will be referred as ineffective regions collectively). For this purpose, we analyze the publicly available and in vivo validated plant sgRNAs data using different statistical tests. We show that nucleotide preference at position near PAM proximal region, GC content in PAM proximal seed region, intact RAR and 3rd stem loop secondary structures, and free accessibility of nucleotides in seed region and tracrRNA

**Table 1 Different plant-specific computational tools for the prediction of sgRNAs.**

| Computational tool | Organism | Off-target prediction/ model or scoring system | Cleavage efficacy/ model or scoring system | Web server address | Reference |
|---|---|---|---|---|---|
| **CRISPR-PLNAT** Version1 | Plants | Yes/*Hsu et al. (2013), Mali et al. (2013), Pattanayak et al. (2013), Li et al. (2013), Nekrasov et al. (2013), Shan et al. (2013), Xie & Yang (2013)* | No | https://www.genome.arizona.edu/crispr/ | *Xie, Zhang & Yang (2014)* (10.1093/mp/ssu009) |
| **CRISPR-PLNAT** Version2 | Plants | Yes/*Minkenberg et al. (2018)* | No | https://www.genome.arizona.edu/crispr2/ | *Minkenberg et al. (2018)* (10.1111/pbi.13025) |
| **CRISPR-P** | Plants | Yes/*Hsu et al. (2013)* | No | http://cbi.hzau.edu.cn/crispr/ | *Lei et al. (2014)* (10.1093/mp/ssu044) |
| **CRISPR-P 2.0** | Plants | Yes/*Doench et al. (2016)* | Yes/*Doench et al. (2014), Bae et al. (2014), Ren et al. (2014), Liang et al. (2016), Lorenz et al. (2016)* | http://cbi.hzau.edu.cn/CRISPR2/ | *Liu et al. (2017)* (10.1016/j.molp.2017.01.003) |
| **CGAT** | Plants | Yes/sequence identity | Yes/*Ren et al. (2014)* | http://cbc.gdcb.iastate.edu/cgat/ | *Brazelton et al. (2015)* (10.1080/21645698.2015.1137690) |
| **CRISPR-GE** | Plant and non-plant organisms | Yes/*Doench et al. (2016)* | Yes/*Ma et al. (2015)* | http://skl.scau.edu.cn/ | *Xie et al. (2017)* (10.1016/j.molp.2017.06.004) |
| **WheatCRISPR** | Wheat | Yes/*Doench et al. (2016)* | Yes/*Doench et al. (2016)* | http://crispr.bioinfo.nrc.ca/WheatCrispr/ | *Cram et al. (2019)* (10.1186/s12870-019-2097-z) |
region of sgRNAs are the most important factors associated with sgRNAs high on-target cleavage efficacy against ineffective regions of *A. thaliana* genome.

## MATERIALS & METHODS

### Retrieval of gRNA sequences

A total of 106 gRNA sequences targeting 53 loci located on different regions of all 5 chromosomes of *A. thaliana* were retrieved from a study carried by *Wu et al. (2018)*. To maintain uniformity and to minimize the possible effects of the backbone/scaffolding region and/or other components (*Hsu et al., 2013*; *Bortesi et al., 2016*), we selected gRNAs from a single study. The target site locations of these gRNAs were determined using the Seqviewer tool available at The Arabidopsis Information Resource (TAIR) database (https://www.arabidopsis.org/). The target sites for these gRNAs were located mainly in regions like 5′ UTRs, 3′ UTRs, introns, intron-exon junctions, near C- and N-terminal ends, exons of either target genes or flanking genes, intergenic regions, and long non-coding RNAs (lncRNAs). The gRNAs were selected based on their target gene(s) knockout ability. The redundant gRNAs and those whose target sites could not qualify as ineffective regions were removed from the dataset. Based on the selection criteria, a total of 58 gRNAs (62%) were determined as highly efficient and contained gRNAs of two different lengths i.e., 19 bp and 20 bp (*Fu et al., 2014*). The base composition was determined using the WebLogo server (https://weblogo.berkeley.edu/logo.cgi), while the observed deletion frequencies for the target genes were taken as cleavage efficiencies of their respective gRNAs.

### Secondary structure prediction and statistical analysis

The prediction of secondary structures of gRNAs and sgRNAs (containing gRNA and scaffolding region) were carried out using RNAfold web server (http://rna.tbi.univie.ac.at/cgi-bin/RNAWebSuite/RNAfold.cgi) available with Vienna RNA software package (*Hofacker, 2003*) accessible at (http://rna.tbi.univie.ac.at/). The software predicts RNA secondary structures based on minimum free energy (MFE) using the Zuker and Stiegler algorithm (*Zuker & Stiegler, 1981*), whereas base pairing probabilities are calculated utilizing the partition algorithm of John McCaskill (*McCaskill, 1990*). Before secondary structure prediction, an additional "G" used for enhancing transcription from U6 promoter (*Wu et al., 2018*) was appended to the sequences. Different statistical tests like Chi-Square, Kruskal-Wallis, and Wilcoxon were employed for determining the features significantly associated with gRNAs high on-target cleavage efficacies, whereas Spearman's rho and Pearson correlation tests were used for inferring the relationship or association of features. The level of significance was taken as <0.05. Chi-Square tests were applied using MS Excel, while all other tests were performed by SPSS software (IBM SPSS Statistics for Windows, Version 21.0; IBM Corp., Armonk, NY, USA). The boxplots were drawn using ggplot2, dplyr, and ggpubr packages in RStudio software.

## RESULTS

### Sequence analysis of gRNAs

The effectiveness of gRNAs in causing on-target editing is of paramount importance in CRISPR-Cas9 mediating genome editing. For determining sgRNAs various features responsible for their high on-target cleavage efficiency, we selected experimentally validated highly efficient gRNAs targeting several ineffective regions of DNA. The selected gRNAs along with their target genes IDs, target site sequences, PAM sequences, strand localization, target site annotation, gRNA sequences, GC content, sgRNA sequences, gRNA, and sgRNA secondary structures with their corresponding MFE values are mentioned in Table S1. The secondary structure of sgRNAs used in the study is shown in Fig. 1. For finding significant features associated with gRNAs high on-target cleavage efficacy, we applied various statistical tests. First, the nucleotide base preference of gRNAs was determined to see if nucleotide base preference is responsible for their on-target cleavage efficacy. Sequence logo created for this purpose revealed a high frequency of thymine at positions 1, 3, 5, 18, and 19, whereas guanine at position 20 along with the frequency change for other nucleotide bases (Fig. 2). Next, we wanted to know if these changes in base frequencies at specific positions have some statistical significance or occurred by chance, so we constructed a frequency table for each position and applied the Chi-Square test (Table 2). The Chi-Square test analysis revealed a significant change in base frequency at position 19 ($p$-value = 8.6E−03). Next, to analyze whether GC content has any impact on activities, we first determined the GC content percentage of full-length gRNAs. Though we could not find any significant change overall ($p$-value = 1.2E−01) but we observed that gRNAs with GC content in ranges of 0–40% and 41–55% showed better cleavage activity as compared to gRNAs with GC content 56–100% (Fig. 3A). As we observed that GC content variation tends to impact the activity, therefore, next we divided the gRNA sequences into sections of different lengths while moving from PAM proximal region to distal region to see if GC content variation only affects the gRNA sub-regions. Since the seed region of gRNA is imperative for the activity, therefore, we calculated the GC content percentage of the PAM proximal seed region (1–12 nt) and the PAM distal region (13–20 nt). In the case of the PAM proximal seed region (1–12 nt), the GC content positively and significantly impacted the cleavage efficacy ($p$-value = 3.2E−02). The comparison of groups demonstrated that significant difference is associated with medium (35–55%) and high (56–100%) GC content groups and cleavage activities remarkably decreased with increasing GC content (Fig. 3B). Regarding PAM distal region (13–20 nt), an overall significant change could not be found ($p$-value = 4.3E−01). However, the cleavage efficacy was positively influenced by increased GC content i.e., 56–100% and 31–55% >0–30% (Fig. 3C). Next, we performed complete tilling with a window size of 5 nucleotides while moving one nucleotide from PAM proximal to distal region across the entire gRNA sequence to see if narrowing down can provide further insight. The 5 nucleotide-wide window tilling could not reveal any significant difference overall ($p$-values range = 0.88–0.06; Figs. S1A–S1P). Further, we determined the impact of the same di and tri contiguous bases on gRNAs on-target editing efficiency. We could not observe any significant effect of dinuceosides

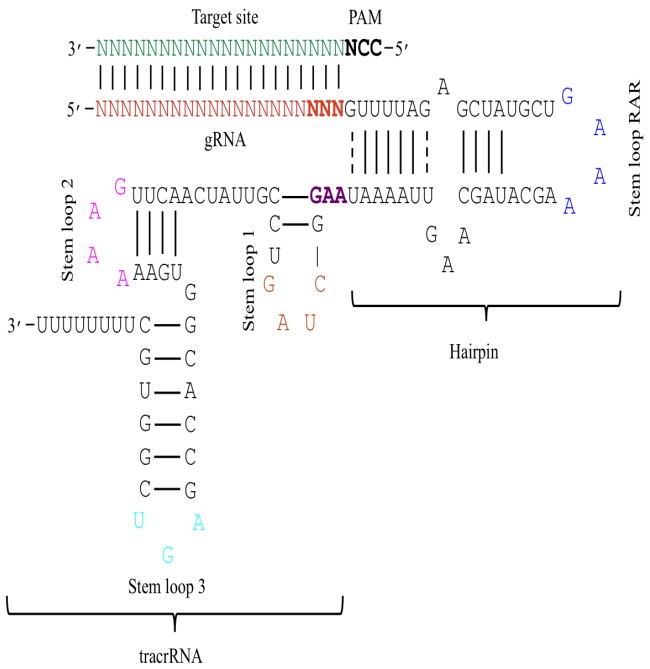

**Figure 1** **The schematic demonstration of the sgRNA secondary structure.** The figure shows that the sgRNA consists of two main parts i.e., gRNA and tracrRNA that are connected through a hairpin-like structure. The tracrRNA and the hairpin-like structure constitute the scaffolding region of the sgRNA. The presence of N (red colour) at 5′ end denotes 20 bp long gRNA sequence that base pairs with the complementary sequence of target DNA (green colour). The bold black colour represents the PAM site that is present adjacent to the target site. Moreover, the secondary structure of sgRNA is characterised by the presence of several secondary structural elements like stem loop RAR (blue colour), stem loop 1 (orange colour), stem loop 2 (pink colour), and stem loop 3 (cyan colour). The last three bases (bold red) of gRNA and the first three bases of tracrRNA (bold purple) mark important nucleotides. The solid lines represent Waston-Crick base pairing, while dashed lines depict non-Waston-Crick or Wobble base pairing.

($p$-values range = 0.97–0.2) and trinucleosides ($p$-values range = 1.0–0.45) on the activity. However, we found that gRNAs with two dinucleosides AA and TT (Figs. 4A–4B) showed non-significantly ($p$-values = 4.4E−01 and 2.0E−01, respectively) enhanced activity compared to gRNAs with two GG and CC where their depletion resulted in more efficient gRNAs (Figs. 4C and 4D). In the case of trinucleosides, gRNAs with depleted trinucloesides (i.e., AAA, GGG, and CCC) positively influenced the cleavage efficacy except for those gRNAs where the presence of one trinucleoside (TTT) or the absence did not make any difference (Fig. 5). Besides determining gRNAs sequence features, we wanted to ascertain if PAM variable nucleotide (VN) and gRNA target DNA strand have any impact on cleavage efficacy. Our analysis showed no significant influence of these features on gRNA cleavage activity (Figs. 6A and 6B).

## Structural features analysis

We manually analyzed sgRNAs secondary structures for the determination of the differences in availability of bases at seed regions (18–20 bp in case of 20 bp long gRNAs and 19–21 bp in case of 21 bp long gRNAs) and tracrRNA regions (59–61 bp and 60–62 bp in case

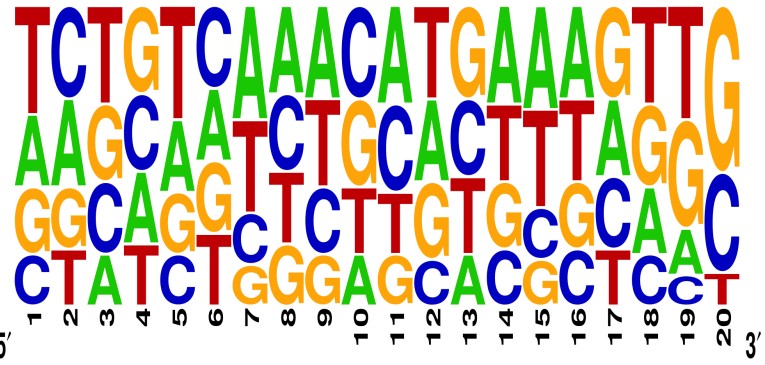

**Figure 2 Sequence logo describing nucleotide preferences in gRNAs targeting ineffective DNA regions.** The figure represents the logo of nucleotide preferences in gRNA sequences and the height of the nucleotide describes its frequency of occurrence at a particular position.

**Table 2 Frequency table showing nucleotide base frequencies at each position of gRNAs.**

| Base position | gRNAs nucleotide base frequency | | | | p-value |
|---|---|---|---|---|---|
| | A | G | C | T | |
| 1 | 14 | 13 | 10 | 21 | 5.2E−01 |
| 2 | 17 | 12 | 18 | 11 | 2.3E−01 |
| 3 | 10 | 15 | 14 | 19 | 5.0E−01 |
| 4 | 14 | 17 | 15 | 12 | 5.5E−01 |
| 5 | 14 | 12 | 10 | 22 | 3.6E−01 |
| 6 | 14 | 14 | 16 | 14 | 7.7E−01 |
| 7 | 22 | 8 | 10 | 18 | 8.3E−02 |
| 8 | 17 | 12 | 15 | 14 | 7.5E−01 |
| 9 | 18 | 10 | 13 | 17 | 6.2E−01 |
| 10 | 10 | 17 | 18 | 13 | 1.7E−01 |
| 11 | 19 | 10 | 17 | 12 | 2.0E−01 |
| 12 | 16 | 15 | 9 | 18 | 6.8E−01 |
| 13 | 10 | 18 | 15 | 15 | 3.5E−01 |
| 14 | 19 | 13 | 11 | 15 | 6.9E−01 |
| 15 | 20 | 9 | 10 | 19 | 2.0E−01 |
| 16 | 18 | 13 | 11 | 16 | 8.2E−01 |
| 17 | 15 | 18 | 14 | 11 | 4.1E−01 |
| 18 | 13 | 16 | 10 | 19 | 6.4E−01 |
| 19 | 9 | 21 | 6 | 22 | 8.6E−03 |
| 20 | 0 | 5 | 3 | 1 | 6.4E−02 |

of 20 and 21 bp long gRNAs, respectively) that can contribute to the on-target cleavage efficacy. We found significant changes at positions 19 (p-value = 6.3E−06), 20 (p-value = 2.4E−03), 59 (p-value = 4.0E−04), 60 (p-value = 3.5E−06), 61 (p-value = 1.7E−05), and 62 (p-value = 5.3E−04) (Table 3). Also, the secondary structures of sgRNAs were analyzed for the presence of intact stem loop elements. We found that stem loop 2 was
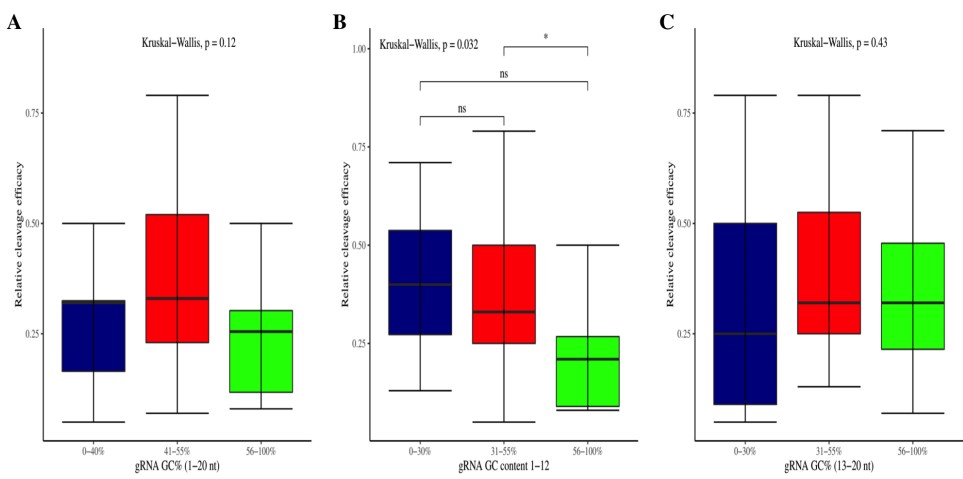

**Figure 3** **Correlation of GC content and cleavage efficacy.** (A) Analysis of gRNAs full-length GC content and cleavage efficacy. No significant difference overall. (B) The GC content of the PAM proximal seed region (1–12 nt) significantly affects the cleavage efficacy. (C) No significant effect of GC content on efficacy within PAM distal region (13–20 nt). Kruskal-Wallis tests are indicated. ns and *, indicate non-significant and significant at 5% probability level, respectively.

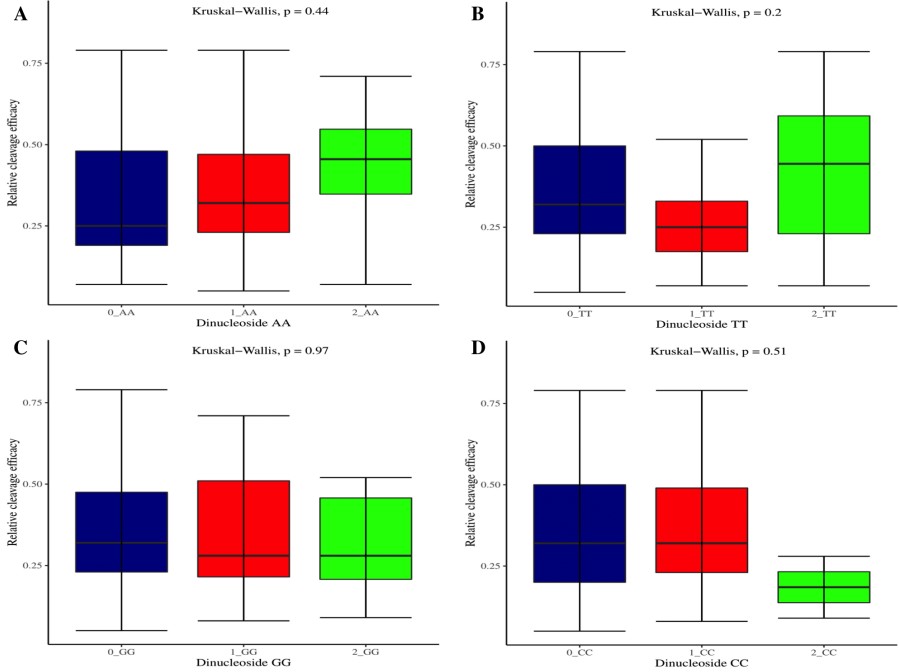

**Figure 4** **Analysis of same dinucleosides impact on cleavage efficacy.** (A–D) Overall no significant effect of same dinucleosides on cleavage efficacy. 0 = gRNAs without dinucleosides, 1 = gRNAs with one dinucleoside and 2 = gRNAs with two dinucleosides. The level of significance is tested with the Kruskal-Wallis test.

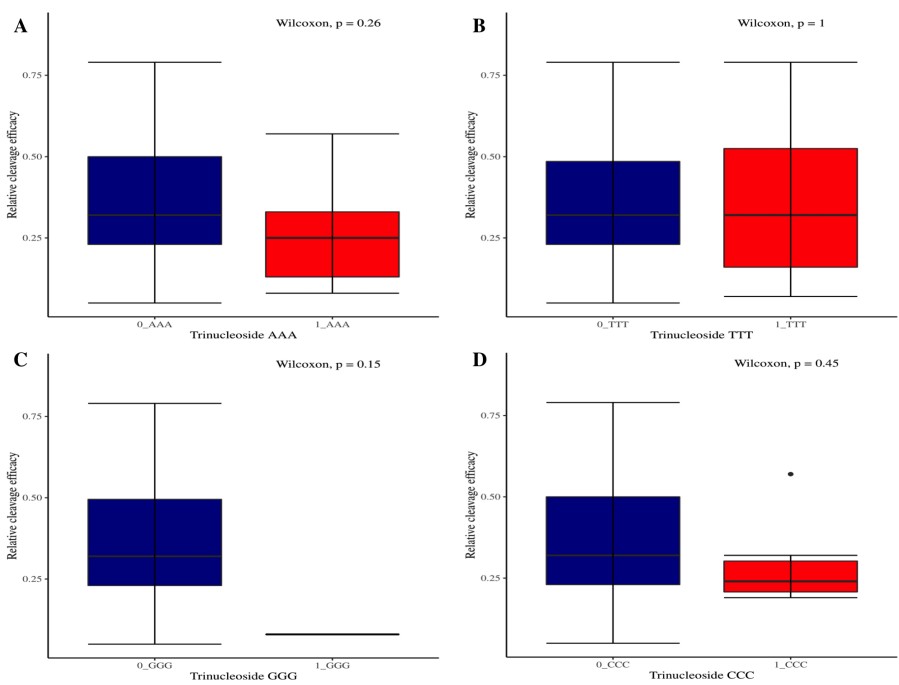

**Figure 5 The correlation of same trinucleosides with cleavage efficacy.** (A–D) The Wilcoxon test shows no significant impact of same trinucleosides on sgRNAs activity. 0 = gRNAs without trinucleoside repeats and 1 = gRNAs with one trinucleoside repeat.

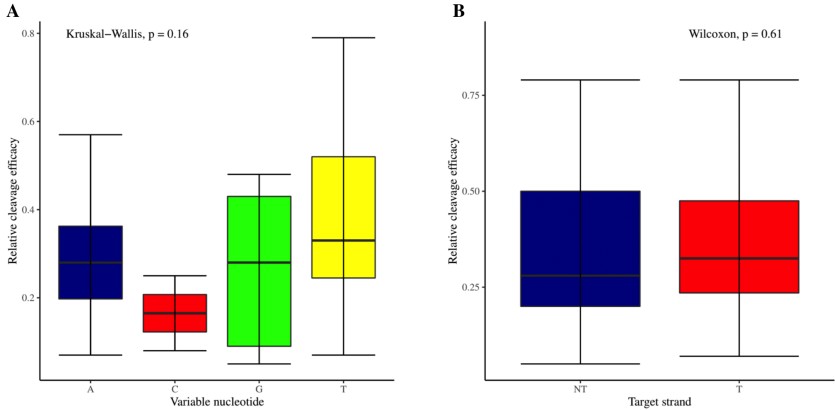

**Figure 6 Analysis of variable nucleotide and target DNA strand impact on cleavage efficacy.** (A) Although no overall significant difference in the usage of PAM variable nucleotides on efficacy is ascertained as indicated by Kruskal-Wallis test. However, the variable nucleotides T (0.3968) and A (0.3031) show better cleavage efficacy than G (0.266) and C (0.165). (B) No overall significant difference as indicated by the Wilcoxon test, however gRNAs targeting transcribed strand show better efficacy compared to those targeting non-transcribed strand.

absent, stem loop RAR and stem loop 3 were present in every sgRNA sequence, while only 5% of the sgRNAs had stem loop 1 structure. Additionally, to determine the influence of

**Table 3  The significantly free accessible nucleotides in the seed region and tracrRNA region of the sgRNAs.**

| | Seed region | | tracrRNA region | | | |
|---|---|---|---|---|---|---|
| Nucleotide position | 19 | 20 | 59 | 60 | 61 | 62 |
| $p$-value | 6.3E−06 | 2.4E−03 | 4.0E−04 | 3.5E−06 | 1.7E−05 | 5.3E−04 |

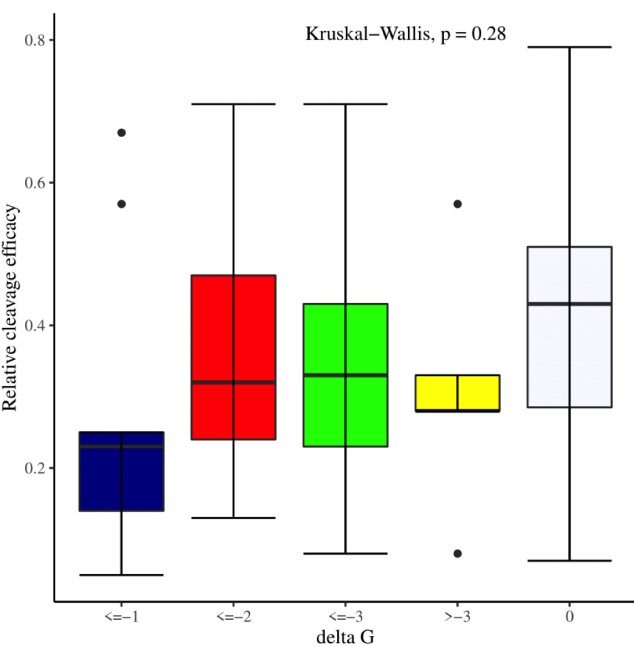

**Figure 7  Effect of gRNAs self-folding free energies (ΔG) on cleavage efficacy.** The Kruskal-Wallis test shows no significant impact of gRNAs ΔG on efficacy overall, however gRNAs with ΔG values 0 demonstrate better efficacy.

secondary structure stability of gRNAs on their cleavage efficacy, we divided the gRNAs secondary structure ΔG values into different stability groups. However, we could not find any statistically significant difference that can relate the gRNAs structure internalization stabilities with their efficacies ($p$-value = 2.8E−01; Fig. 7).

## Association of cleavage efficacy with sgRNAs features

We carried out Spearman's rho correlation and Pearson correlation tests for determining the association of gRNAs full-length GC content, ΔG of gRNAs, and ΔG of sgRNAs with cleavage efficacies, respectively. In the case of GC content, a negative and a very weak association was seen with cleavage efficacy ($r = -0.123$). For gRNAs ΔG and cleavage efficiency, a positive and a very weak relationship was observed ($r = 0.089$), while in the case of sgRNA ΔG, a positive and a very weak association with cleavage efficacy was observed ($r = 0.194$). However, in all the aforementioned cases the statistically significant relationship could not be observed between the variables ($p$-values = 3.91E−01, 5.33E−01, and 1.71E−01, respectively).

## DISCUSSION

The current study identifies various sequence and structural features of sgRNAs as important determinants of their high on-target cleavage efficacy against ineffective regions of genomic DNA that are of immense importance in functional genomics studies. Keeping in mind the importance of ineffective regions, the study targets DNA regions that are excluded from the sgRNA design model (*Doench et al., 2014*) along with other non-coding genomic regions for determining the various sequence and structural features that affect sgRNAs on-target cleavage efficacies. For this purpose and to comprehend the in vivo high on-target cleavage efficacy of sgRNAs, we used publicly available and in vivo validated plant sgRNAs targeting various ineffective regions. Next, to establish the criteria, which can demonstrate their high on-target cleavage efficacies, we applied different statistical tests. The analysis revealed a statistically significant difference at position 19, which constitutes the 3′ end of gRNAs. The position 19 is present adjacent to PAM in 19 bp long sgRNAs and agrees with previous observations (*Doench et al., 2014*; *Wang et al., 2014*; *Xu et al., 2015*; *Liu et al., 2016*). However, in contrast to previous observations that reported guanine or cytosine as a preferred base at a position adjacent to PAM, our study showed thymine is dominating at this position, while at other positions we did not observe any significant change (Table 2). The dominance of thymine reflects the AT-rich nature of non-coding regions. In 20 bp long gRNAs, we could not find any significant change at position 20, adjacent to PAM, which might be due to the absence of data and/or their small sample size at this position. The gRNA GC content was shown to have an effect on sgRNAs activities with low or high GC content resulting in the generation of inefficient sgRNAs (*Doench et al., 2014*). Our results showed no statistically significant difference in the GC content of full-length gRNAs (1-20 nt) (Fig. 3A). However, the analysis of GC content of gRNAs split sections showed that the GC content of PAM proximal seed region (1–12 nt) impacts the cleavage efficacy significantly and increasing GC content significantly decreases gRNAs activity (Fig. 3B) and is in disagreement with the former studies that could not find GC content significant impact in this region (*Ren et al., 2014*; *Labuhn et al., 2018*). The presence of the same contiguous bases (TTT, GGG, and GG) may interfere with gRNA transcription or affect their editing efficacies (*Wong, Liu & Wang, 2015*). The analysis of our dataset could not reveal any significant correlation of di- and trinucleosides with the efficacy (Figs. 4 and 5), however the observed non-significant increase in gRNAs efficacy with dinucleosides AA, TT, and trinucleosides TTT seems associated with the nature of non-coding regions. In our dataset, the analysis of PAM variable nucleotide and target DNA strand, taken as a function of gRNAs activity, showed no statistically significant impact on their cleavage efficacy (Figs. 6A–6B), which were in contrast to the previous observations (*Doench et al., 2014*; *Wang et al., 2014*). The free accessibility of the last three bases of the seed region and the first three bases (AAG) of the tracrRNA region is imperative for on-target cleavage efficiency (*Wong, Liu & Wang, 2015*). Our results are in agreement with the aforementioned observations as we found significant differences at these positions (Table 3). Different stem loop elements in secondary structures like RAR, 2nd and 3rd stem loops were shown to be associated with plant sgRNAs on-target efficiency (*Ma et al., 2015*;

*Liang et al., 2016*). The results showed that the presence of intact RAR and 3rd stem loop structures are important for their on-target cleavage efficacy and the absence of 2nd stem loop element indicates that this secondary structural element does not have any impact on their cleavage efficacy against ineffective genomic regions. Previous studies showed that energetically stable gRNAs secondary structures are responsible for cleavage inefficiencies (*Wong, Liu & Wang, 2015*; *Thyme et al., 2016*; *Jensen et al., 2017*), which were in contrast to our results, as we could not find statistically meaningful difference in self-folding free energies of gRNAs across different stability groups (Fig. 7). The gRNAs GC content and ΔG of gRNAs were shown to significantly impact the cleavage efficacy (*Ma et al., 2015*; *Jensen et al., 2017*). Our results demonstrated a very weak relationship of these parameters with cleavage efficiencies. However, the significance of the observed relationships could not be established. Further, our results showed no association of ΔG of sgRNAs with the efficacy, which is in agreement with previous findings (*Jensen et al., 2017*). Interestingly, we also found some gRNAs in the dataset, which once were non-functional and became functional against the same target sites upon swapping and vice versa, indicating that some other extrinsic factors besides sequence and structure features are also working for determining their functionality (*Durr et al., 2018*).

The results of our study demonstrate that the sgRNAs targeting plant ineffective regions are different in various parameters from the sgRNAs designed against protein-coding regions of the mammalian genomes. This indicates the need for designing high throughput CRISPR screening studies considering the ineffective regions in addition to the whole genomic landscape in plants. The difference in sgRNAs activities against plants and mammalians genomes was also demonstrated during the formation of design criteria for efficient sgRNAs prediction using in vivo validated plant sgRNAs targeting different genes across different plants (*Liang et al., 2016*). Despite demonstrating different features associated with sgRNAs high on-target cleavage efficacy against ineffective genomic regions, the experimental validation of these results is required.

## CONCLUSIONS

In conclusion, our study demonstrates the features and parameters governing sgRNAs with high on-target efficacy against otherwise ineffective regions of the *A. thaliana* genome. Our findings illustrate that the ineffective regions of the genome are equally important to consider while designing sgRNAs prediction models. Moreover, we show that plant sgRNAs targeting various ineffective regions of DNA do not strictly follow the parameters designed for protein-coding regions, which are implemented in various sgRNAs design tools. These results indicate the requirement of designing plant genome wide CRISPR screening studies considering the entire genomic context for the rapid prediction of efficient sgRNAs. In this regard, our study can serve as a paradigm for the comprehensive analysis of hundreds of sgRNAs sequences for inferring highly meaningful and statistically significant features

for the development of a cost- and time-efficient plant sgRNAs design tool. The prospects encompass the experimental validation of the outcomes of the study.

### Funding

The authors received no funding for this work.

### Competing Interests

The authors declare there are no competing interests.

### Author Contributions

- Afsheen Malik conceived and designed the experiments, performed the experiments, analyzed the data, prepared figures and/or tables, authored or reviewed drafts of the paper, and approved the final draft.
- Alvina Gul, Faiza Munir, Rabia Amir, Hadi Alipour, Mustafeez Mujtaba Babar, Syeda Marriam Bakhtiar, Rehan Zafar Paracha, Zoya Khalid and Muhammad Qasim Hayat analyzed the data, authored or reviewed drafts of the paper, and approved the final draft.

### Data Availability

The raw data is available in the Supplementary Files.

### Supplemental Information

Supplemental information for this article can be found online at http://dx.doi.org/10.7717/peerj.11409#supplemental-information.

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
