# Peer review of "Evaluating the cleavage efficacy of CRISPR-Cas9 sgRNAs targeting ineffective regions of Arabidopsis thaliana genome"

_PeerJ, doi:10.7717/peerj.11409_

## Round 0.1 · original submission · Major Revisions

I agree with the reviewers' comments and the issues they have raised. If you could adequately address these and provide some experimental evidence for your findings, we can reconsider your significantly revised manuscript. It is highly likely that the manuscript will be sent for review again to the same reviewers.

Reviewer 1 ·

Basic reporting

With the exception of minor changes to the text (listed in General Comments), Malik and colleagues have clearly and professionally written this manuscript and provided sufficient figures and data. The introduction and references are appropriate for this study.

Experimental design

Statistical analyses in this study are appropriate and accurately assess some of the features affecting sgRNA cleavage efficiency. This work is relevant as many other studies on sgRNA efficiency determinants focus on mammalian and/or coding regions. As interest in targeting ineffective regions grows, resources such as this manuscript for designing efficient gRNAs will be useful and interesting to the plant science community.

Validity of the findings

A significant concern I have with this study is the interpretation of the data used. A dataset is chosen from Wu et al 2018 in which several pairs of gRNAs are used to remove coding sequences of various genes in Arabidopsis thaliana. Malik and colleagues then remove gRNAs targeting coding regions and split the remaining gRNAs into two categories based on their ability to remove the coding sequence in Arabidopsis, as reported by Wu et al 2018. This binary functional/non-functional split is then used for the remainder of the analysis. Interpreting the data in this way results in two issues. First, Wu and colleagues report various gene deletion efficiencies for the gRNA pair. While it’s true that any deletion is an indication that the gRNA is functional, the relative efficiency of the gRNA pair could also be considered in the analysis. Second, and of major concern, is assigning gRNAs to the non-functional group assumes that both gRNAs must be ineffective at cleaving DNA. This is untrue as deletion events, as reported in Wu et al 2018, can only occur when both gRNAs are functional and have relatively the same cleavage efficiency. It’s possible that in many cases, for each gRNA pair in which a deletion is not recovered (and thus assigned to the non-functional gRNA group in this study), one of the two gRNAs is indeed functional. This results in an unknown number of gRNAs being incorrectly assigned to the non-functional category and impacting downstream statistical analysis used for the rest of the study.

Other considerations:

As this study examines sgRNA features affecting efficiency when specifically targeting ineffective regions, the study would be strengthened if analysis were also performed in parallel on sgRNAs targeting coding regions to compare differences in features for targeting each of these regions.

The study would be greatly strengthened if the authors also include experimental validation of their findings. The authors acknowledge this.

Additional comments

I found the research question relevant and an often-undervalued component of CRISPR/Cas9 gene editing. The interpretation of the initial dataset is incorrect, but I would encourage the authors to examine the same research question with an appropriate dataset. Data in which each gRNA cleavage efficiency is reported individually rather than as a pair should allow effective statistical analysis of the sgRNA features affecting cleavage efficiency.

Line 101- either remove “much” or replace with “very”

Line 139-140- wording is confusing, in particular “potentially” and “inactive”

Line 154- “sites” to “site”

Line 157- remove “and”, not the end of the list

Line 205- replace “either” with “whether”

Line 211- Include p-value of significance in main text

Line 213- replace “high” with “higher”, comparing two groups

Line 220- Include p-value of significance in main text

Line 252- replace “out” with “any”

Line 287- remove “Though”

Reviewer 2 ·

Basic reporting

I believe. that there is data which contributes CRISPR community. I do have two issues with the MS:
1. Please remove too old references unless they are not pioneer,

2. English editing is a must and there is typos and should be checked.

3. The following recent review/research article would be good to add/discuss and compare with this research
- CRISPR/Cas9 in plants: at play in the genome and at work for crop improvement. Briefings in functional genomics 17 (5), 319-328 16 2018
-CRISPR/Cas9 genome editing in wheat. Functional & integrative genomics 18 (1), 31-41 86 2018

4. Figure 2 can be replaced with a different one. Is it possible show it differently?

Experimental design

This is good to show in Arabidopsis.

Validity of the findings

It is done.

Additional comments

I believe. that there is data which contributes CRISPR community. I do have two issues with the MS and listed above.

---

## Round 0.2 · Major Revisions

We request the authors to address the issues raised by reviewer #1.

Reviewer 1 ·

Basic reporting

The text is now clear and well written.

Experimental design

My position has not changed. The statistical analysis is appropriate for the original question and work is relevant.

Validity of the findings

I agree with the authors rational for the study, that targeting ineffective regions may produce lower editing efficiencies.
I also agree with the author’s rebuttal that dual gRNAs may be recommended to enhance targeting efficiency, but this study is not examining pairs of gRNAs.
The dataset and conclusions would be appropriate if the statistical analysis considered sequence features of pairs of gRNAs. As is stated in their rebuttal letter, “for evaluating their features that collectively are responsible”. It is indeed possible that there are sequence features within a gRNA pair that impact editing efficiency but the authors are examining sequence features of individual gRNAs. The statistical analysis treats gRNAs as individuals once they are assigned to “functional” or “non-functional” groups.
Due to this fact, I still have issue with the dataset used in this study. In the authors rebuttal letter, they mention contacting the corresponding author of Wu et al., 2018, from which the data set was obtained, and received the response “they were functional but not always highly efficient” regarding gRNAs that did not produce deletions. This suggests some gRNAs that did not produce deletions were indeed highly efficient. Inappropriately assigning these individual gRNAs to the “nonfunctional” group would then affect the statistical analysis.
I believe it’s likely that many of the gRNAs are correctly assigned to functional and non-functional groups but even if a few “non-functional” gRNAs could produce high individual targeting efficiencies, the final conclusions of the statistical analysis could be compromised.
The authors have adequately addressed all of my other comments.

Additional comments

None

Reviewer 2 ·

Basic reporting

There revised version seems better.

Experimental design

Looks good

Validity of the findings

There would be Moree validation however this is good enough for the paper.

---

## Round 0.3 · accepted · Accept

The revised manuscript has addressed most of the issues raised by the reviewers.